# Deploying deep learning models on unseen medical imaging using adversarial domain adaptation

Aly A. Valliani[1], Faris F. Gulamali[1], Young Joon Kwon[1], Michael L. Martini[1], Chiatse Wang[2], Douglas Kondziolka[3,4], Viola J. Chen[5], Weichung Wang[2,6], Anthony B. Costa[7], Eric K. Oermann[3,8]*

1 Department of Neurosurgery, Mount Sinai Health System, New York, NY, United States of America, 2 Data Science Degree Program, National Taiwan University, Taipei, Taiwan, 3 Department of Neurosurgery, New York University Langone Medical Center, New York, NY, United States of America, 4 Department of Radiation Oncology, New York University Langone Medical Center, New York, NY, United States of America, 5 Oncology Early Development, Merck Co., Inc, Kenilworth, NJ, United States of America, 6 Institute of Applied Mathematical Sciences, National Taiwan University, Taipei, Taiwan, 7 NVIDIA, Santa Clara, CA, United States of America, 8 Department of Radiology, New York University Langone Medical Center, New York, NY, United States of America

* eric.oermann@nyulangone.org

**Data Availability Statement:** All data were approved for research use by each institution's respective Institutional Review Board. Data utilized in this study are publicly available and accessible in

## Abstract

The fundamental challenge in machine learning is ensuring that trained models generalize well to unseen data. We developed a general technique for ameliorating the effect of dataset shift using generative adversarial networks (GANs) on a dataset of 149,298 handwritten digits and dataset of 868,549 chest radiographs obtained from four academic medical centers. Efficacy was assessed by comparing area under the curve (AUC) pre- and post-adaptation. On the digit recognition task, the baseline CNN achieved an average internal test AUC of 99.87% (95% CI, 99.87-99.87%), which decreased to an average external test AUC of 91.85% (95% CI, 91.82-91.88%), with an average salvage of 35% from baseline upon adaptation. On the lung pathology classification task, the baseline CNN achieved an average internal test AUC of 78.07% (95% CI, 77.97-78.17%) and an average external test AUC of 71.43% (95% CI, 71.32-71.60%), with a salvage of 25% from baseline upon adaptation. Adversarial domain adaptation leads to improved model performance on radiographic data derived from multiple out-of-sample healthcare populations. This work can be applied to other medical imaging domains to help shape the deployment toolkit of machine learning in medicine.

## 1 Introduction

A major point of failure for machine learning models is lack of generalizability to unseen cases when deployed in production [1]. A major cause of this is dataset shift, when the underlying population (or domain) from which a model's training set is sampled has a different distribution from the population encountered in production [1–3]. This problem for generalizing algorithms is thought to be a major challenge facing autonomous cars, financial systems, and

**Funding:** The author(s) received no specific funding for this work.

many other deep learning systems. For medical models where it is common for datasets to reflect local patient populations and image acquisition methods, this problem is uniquely prevalent [1, 4–6]. The failure of computer-assisted diagnosis for mammography despite its approval by the FDA is one of the most well-known medical cases [6]. The most straightforward way to address this problem is by obtaining information about the external distribution via acquiring labeled data from it. However this is particularly challenging in many fields such as medicine, for example, where data is siloed in healthcare institutions to protect patient privacy, high-quality labeled data is time-consuming to acquire, and requires esoteric knowledge of the field [7, 8]. In medicine, high-quality labels are especially expensive to acquire as they require multiple graders to derive consensus in the context of poor intergrader reliability [9–11]. As such, purely technical approaches belonging to the realm of transfer learning and domain adaptation are promising alternatives. Broadly, algorithms for domain adaptation can be categorized into instance-based or feature-based approaches [12, 13]. Instance-based domain adaptation applies a re-weighting function to reduce the discrepancy between source and target samples whereas feature-based approaches aim to learn a mapping across domains when labeled data is unavailable in the target domain. The latter is the focus of this work.

Prior work has utilized variations of adversarial domain adaptation on a spectrum of different tasks including medical image segmentation, lung nodule detection, prostate MRI segmentation, and federated learning. For example, previous methods have trained on augmented big data in the domains of prostate, left atrial, and left ventricular and shown that augmentation reduces the degradation in performance significantly [14]. In our study, we primarily focus on an in-hospital vs out-of-hospital cohort, rather than differing tasks altogether. A second method has utilized adaptive transition module (ATM) to learn a frequency attention map that can align different domain images in a common frequency domain. By backpropagating with differentiable fast fourier transform, lung nodule detection performance was significantly improved [15]. We do not use a frequency domain, but we anticipate that applying a frequency-based normalization may also improve performance. Shape-aware meta learning utilizes a network that can learn shape compactness and shape smoothness to provide domain-invariant embeddings [16]. Similar to ATMs, shape-aware meta-learning is primarily focused on different objectives rather than learning out-of-sample embeddings. Finally, some methods are able to combine Fourier transforms and shape-aware meta learning, demonstrating improved performance on out-of-sample objectives [17]. In context, our paper focuses on investigating the a priori assumption of dataset shift, and how it can be utilized to improve performance across centers rather than generating a novel machine learning methods to combat domain shift.

We utilize an unsupervised domain adaptation algorithm that relies upon generative adversarial networks—neural networks that compete with one another—to obtain state-of-the-art results across all transformations for a canonical digit recognition task as well as one of the largest medical imaging datasets curated to date. Using this algorithm and datasets, we examine different scenarios for deploying a machine learning model in a medical use-case, analyze points of failure, and demonstrate the efficacy of our technique for maximizing data efficiency. An experimental innovation that we emphasize is refraining from presenting results from a joint test set. Rather we present results from distinct test sets split on our prior expectation of dataset shift (digit source or hospital site in our two cases respectively) and show that this simple change significantly improves our understanding of the problem, particularly in the medical use case where it can be common to test results on pooled multicenter data.

Two categories of data were used for algorithm development and validation. Handwritten digit datasets were used for initial prototyping and proof-of-concept testing, and clinical chest x-ray (CXR) datasets were leveraged to simulate translation into the clinical setting. Our digit

dataset consisted of 149,298 images of digits from three classic populations: MNIST, MNISTM, and USPS [18, 19]. We then applied our work to what, to the author's best knowledge, is the largest dataset of medical imaging to date consisting of 868,549 chest radiographs drawn from 228,258 patients from three academic medical centers within the United States (Beth Israel, Stanford, and the National Institutes of Health Clinical Center) as well as an academic medical center in Spain (San Juan Hospital) [20–23].

There are four primary means of deploying a machine learning model in an environment with distinct populations (Fig 1A). We refer to "internal" (in-population or in-dataset) results as classification results tested on a held-out test set sampled from the same population as the training set. In contrast, we refer to external (out-of-population or out-of-dataset) results as classification results where the model is tested on a held-out test set sampled from a different dataset as the training set (see Materials and methods for further details). An idealized case is to train a model on a local dataset, and then have it perform well externally, out-of-dataset, on multiple different datasets. We developed a technique for generally improving algorithm performance using a purely computational approach involving cycle-consistent adversarial domain adaptation to make data from one dataset mimic that from another dataset. Our system is built upon the generative adversarial framework consisting of a single "generator" deep neural network (DNN) that competes against a "discriminator" DNN in a game to detect forged images (Fig 1B) [24]. The generator is tasked with taking input images from a source domain, and making them appear as if they were sampled from a target domain (Fig 1C). Once we have learned a technique for transferring data between domains, it is possible to train a classification model on any one population and then deploy it on an external one while minimizing the loss in performance. Baseline classification performance is the internal area under the curve (AUC), or the AUC on a held-out test set sampled from the same domain as the one the neural network classifier was originally trained on. Efficacy of domain adaptation is measured by comparing the post-adaptation AUC to the baseline AUC on the same held-out test set. Detailed descriptions of the datasets, model, and training routine can be found in the Materials and Methods.

## 2 Results

### 2.1 Digit recognition

On a standard digit recognition task, we noted an average internal AUC of 99.87% (95% CI, 99.87–99.87%) which decreased to an average external AUC of 91.85% (95% CI, 91.82–91.88%) without adaptation and an average external AUC with adaptation of 94.66% (95% CI, 94.63–94.69%). Adaptation led to a generalized increase in performance with average salvage of approximately 35% post-adaptation as compared to baseline (Fig 2A). Notably, there was a global increase with adaptation across all datasets and, on visual inspection, adapted digits appear to be semantically consistent with their target datasets even across gray-scale and color transformations (Fig 2B and 2C). On average, external testing of locally trained models demonstrates a relative increase in performance of 3.04% (95% CI, 3.01–3.07%, absolute increase of 2.81%) and exhibits state-of-the-art results on this task for domain adaptation with a median salvage of 83.16% of the AUC lost from testing on an external dataset. Domain adaptation worked best on domains that were more similar under visual inspection. For example, adaptations between MNIST and MNISTM yielded significant improvements in classification performance due to the similar baseline character style across the two datasets. Adaptations between MNIST and USPS were similarly efficacious due to transition across grayscale domains whereas adaptations between MNISTM and USPS were less successful given the more difficult task of adaptation across character styles and color domains.

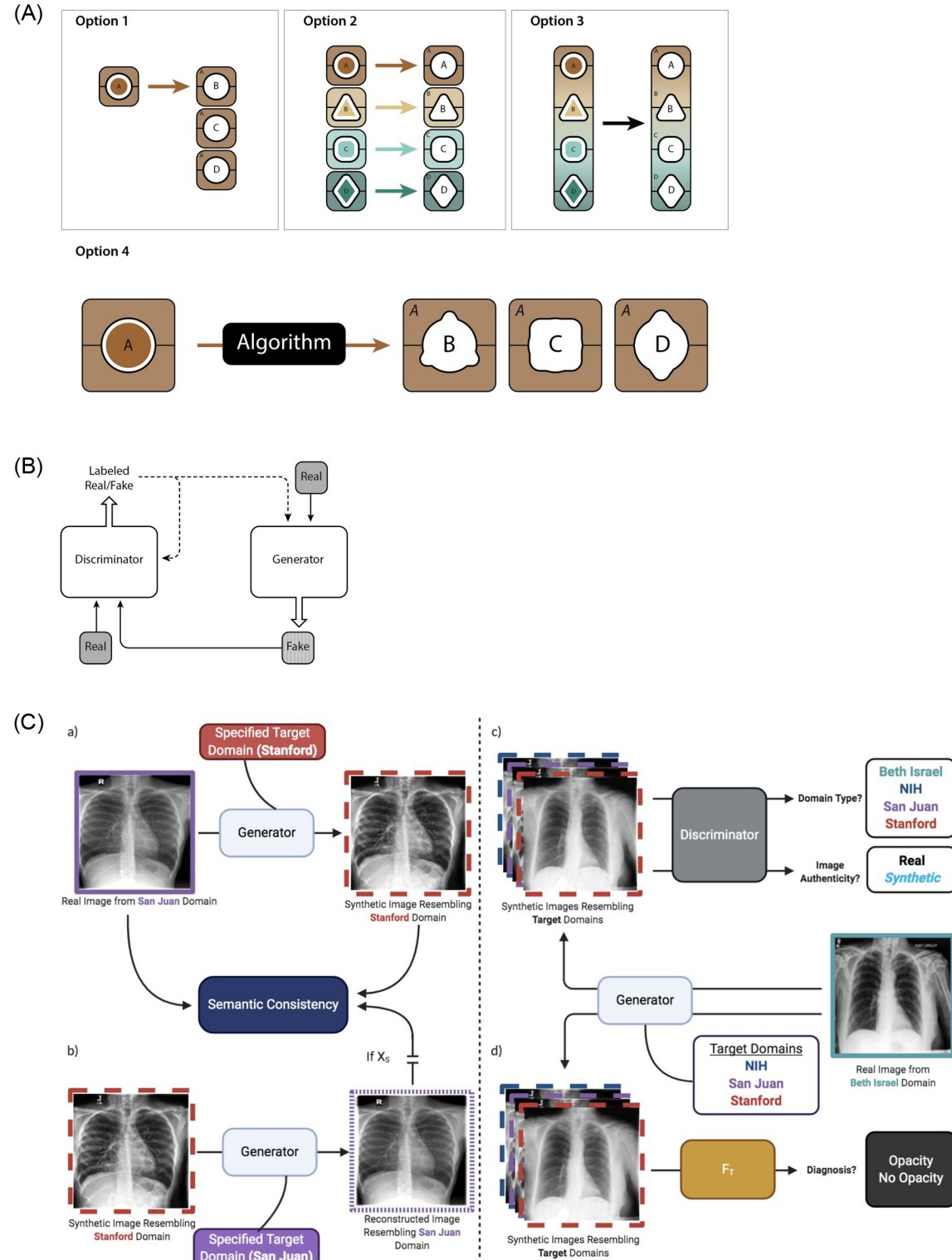

**Fig 1. Machine learning deployment strategies and schematic illustration of the proposed generative adversarial algorithm for domain adaptation.** (A) There are four primary methods by which machine learning models can be deployed in a context with distinct data domains: 1) train a model on one domain and deploy it across multiple distinct domains, 2) train multiple bespoke models that are optimized for deployment on individual domains, 3) train and deploy a single global model on all domains, and 4) train a model on one domain and adapt it through technical means to make it performant on a distinct domain. (B) Generative adversarial networks provide

a technical framework for domain adaptation. A generator translates real data from one domain into fake data that resembles that of a different domain while the discriminator aims to distinguish between the two, which enables the generator to generate realistic-looking data in the target domain. **(C)** Schematic of the proposed algorithm. a) Real data from a source domain is translated by the generator to resemble data from a specified target domain while maintaining underlying semantic qualities of the input image. b) Translated data is reconstructed by the generator to resemble data from the source domain to maintain domain-agnostic image characteristics with a semantic consistency constraint ensuring that reconstructed images maintain the semantic characteristics of the source data. c) The discriminator aims to distinguish between real and synthetic images and identify the domain of input images to constrain the generator to produce realistic-looking synthetic images from a specified domain. d) A target discriminator is fine-tuned on synthetic images to better identify opacity in the target domain.

## 2.2 Chest x-ray classification

Using one of the largest medical imaging datasets to date we tested our technique on a medical imaging problem, identification of opacities on chest x-rays (CXRs), which is of particular relevance due to the present need for using algorithms to rapidly spot pulmonary aberrations from COVID-19 (Materials and Methods: Clinical Taxonomy and Pre-Processing) [25–29]. On our medical dataset we had an average internal test AUC of 78.07% (95% CI, 77.97–78.17%), and an average external AUC of 71.43% (95% CI, 71.32–71.60%) with an average relative performance loss of 8.51% (S3 Table in S1 File). After adaptation we noted an average relative improvement in performance of 2.42% (95% CI, 2.30–2.54%, absolute increase of 1.64%) implying an average salvage of approximately 25% of baseline performance (Fig 3A, S4 Table in S1 File). Specific populations tended to suffer more from dataset shift in the unadapted setting, and ultimately benefit more from adaptation—in this case San Juan with an average relative gain of 6.58% (absolute increase of 4.39%) AUC after adaptation (Fig 3B). For context, achieving this level of improvement without adaptation would require on average an additional 8,213 labeled chest radiographs derived from the target domain. Implementing domain adaptation more broadly would amount to having approximately 5,845 additional labeled images from the deployment dataset (S1 Fig in S1 File). In order to confirm that performance gains were due to adaptation of the underlying data and features rather than an incidental re-calibration, we visually inspected the adapted CXRs and plotted calibration curves confirming that adaptation is acting upon the underlying distribution of features rather than simply recalibrating the models (Fig 3C, S2 Fig in S1 File). For the CXR models there was a median salvage of 20.98% of AUC after adaptation.

## 2.3 Domain spread

The present work displays encouraging and practically useful results across both non-medical and medical datasets for mitigating dataset shift in the challenging case of not having access to labeled data in the target domain. When dealing with easily transported data, this problem can be somewhat obviated by simply localizing the data and training models on a union of the data or utilizing other techniques from transfer learning. Importantly, however, we observe that the performance of global models on pooled data from multiple data sources does not reflect efficacy on individual data domains (Fig 4, S4 and S5 Tables in S1 File). Instead, stratifying assessment by domain allows for examination of the domain spread, or inter-domain variance, as an a priori measure of expected model performance upon deployment. The dramatic reduction in domain spread with increasing amounts of handwritten digits data (0.1% domain spread = 112.06 and 100% domain spread = 0.01) relative to that of CXR data (0.1% domain spread = 26.12 and 100% domain spread = 23.83) suggests that added radiographs may not be sufficient to overcome data shift across hospital sites.

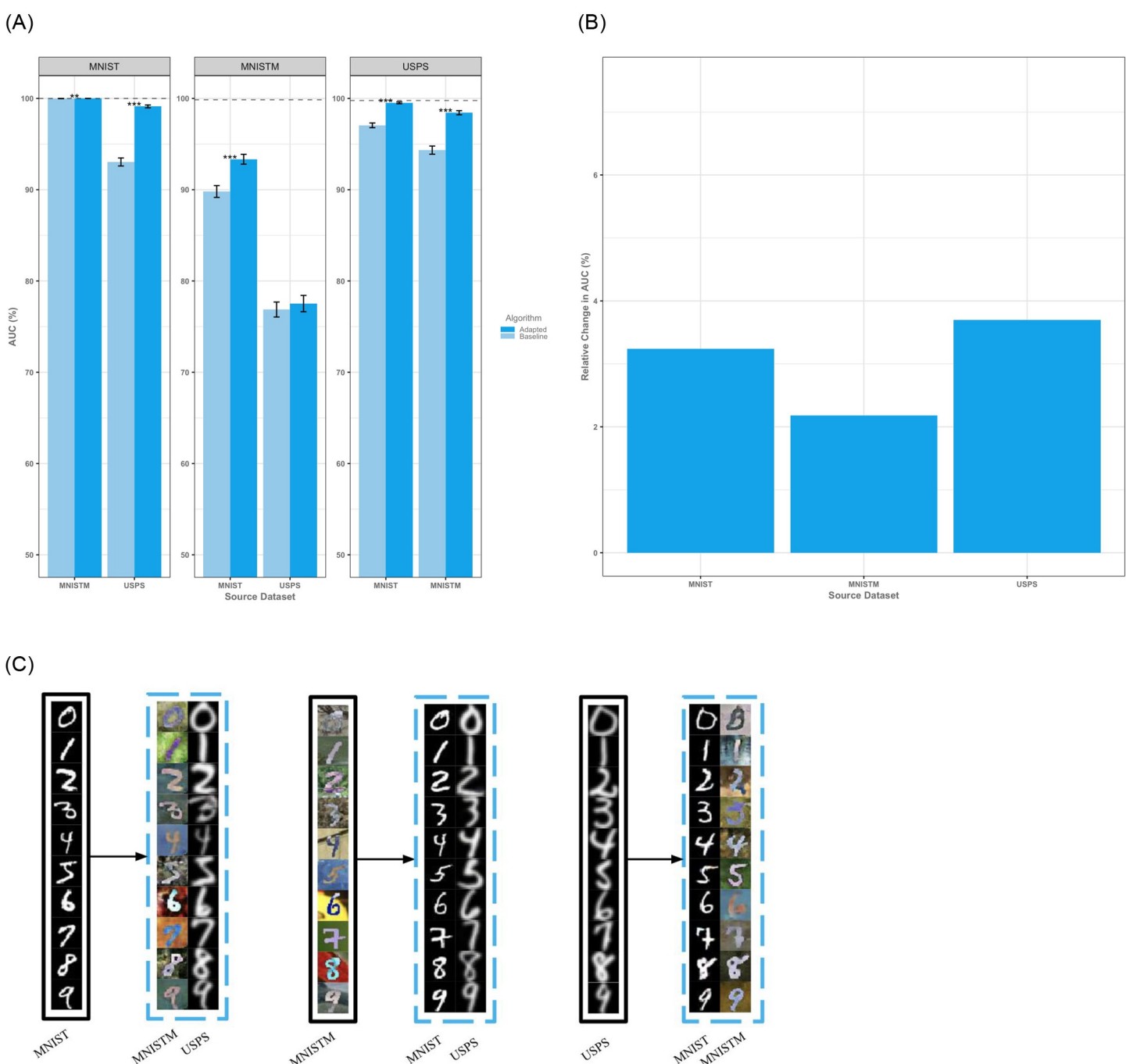

**Fig 2. Results on the digits datasets. (A)** Performance of adapted and baseline algorithms as measured by area under the curve (AUC). Error bars denote standard deviations. Dotted lines represent the theoretical ceiling of AUC on the target test set as obtained by a baseline classifier trained on the target training set. Adaptation leads to a generalized increase in AUC across all source-target pairs with an average salvage of 35% of peak performance. **(B)** Expected relative change in AUC upon adaptation of a source dataset demonstrates a generalized increase in performance across populations. **(C)** In all cases, adaptation transforms input images (bounded by black boxes) to appear stylistically like those in the specified target domain (bounded by blue boxes) while preserving semantic information of images in the source domain.

## 3 Discussion

We built an unsupervised domain adaptation algorithm using GANs that ameliorates dataset shift on a canonical computer vision task and among the largest medical imaging datasets ever

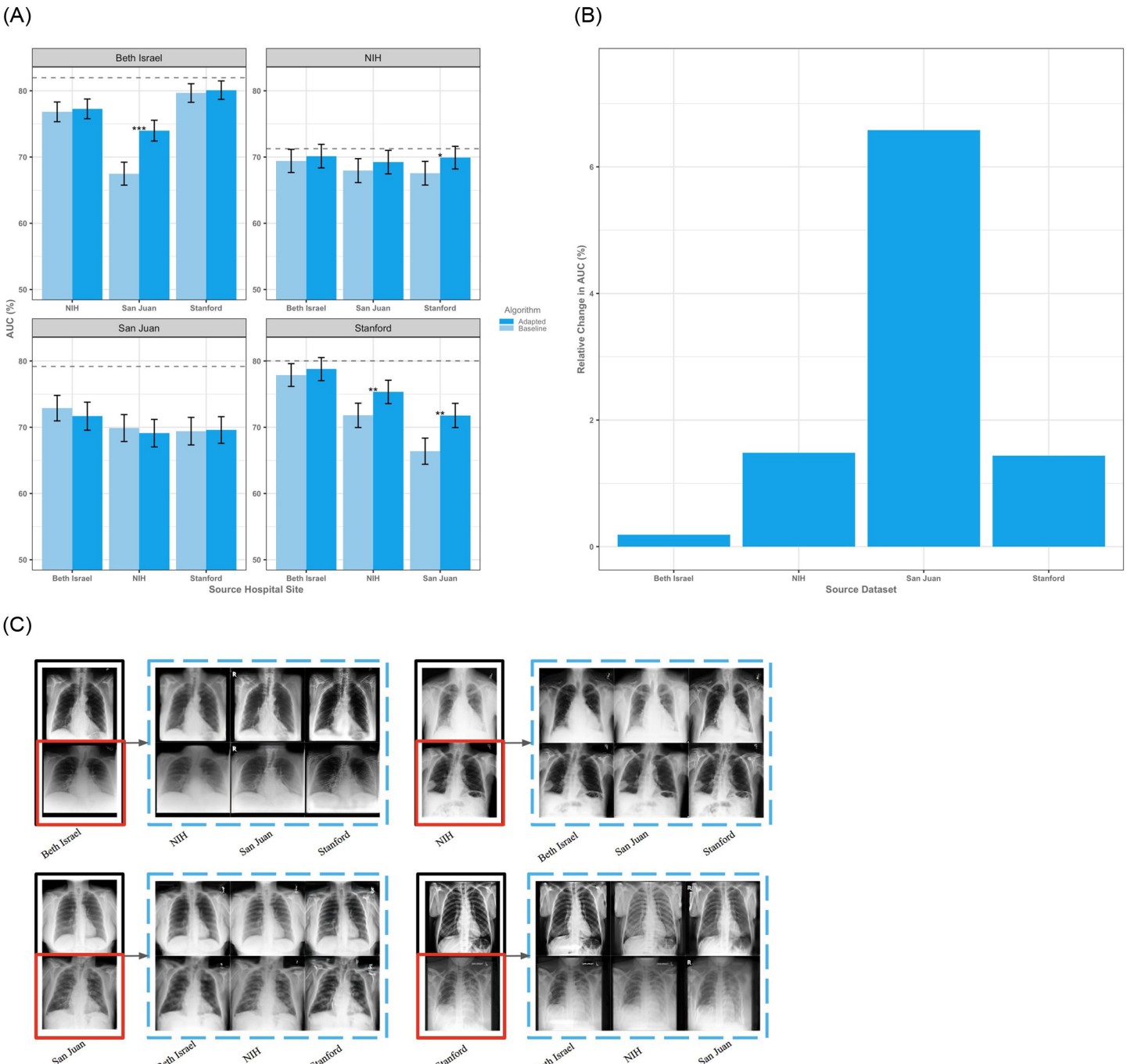

**Fig 3. Results on the chest x-ray datasets. (A)** Performance of adapted and baseline algorithms as measured by area under the curve (AUC). Error bars denote standard deviations. Dotted lines represent the theoretical ceiling of AUC on the target test set as obtained by a baseline classifier trained on the target training set and demonstrate an average salvage of 25% of the baseline performance after adaptation. **(B)** Expected relative change in AUC upon adaptation of a source dataset demonstrates a general improvement in performance across populations. The proposed adaptation technique leads to a generalized increase in AUC on average relative to baseline performance. **(C)** Input images without opacity are bounded by black boxes while those with opacity are bounded by red boxes. Adapted counterparts are bounded by blue boxes.

curated. Our approach extends upon previously described methods by incorporating real-time weak supervision of the generator to enable semantic consistency. Most importantly, unlike previous work, we conduct a comprehensive validation of our algorithm across all adaptation pairs for two very different computer vision tasks. As a proof-of-concept, we show broad

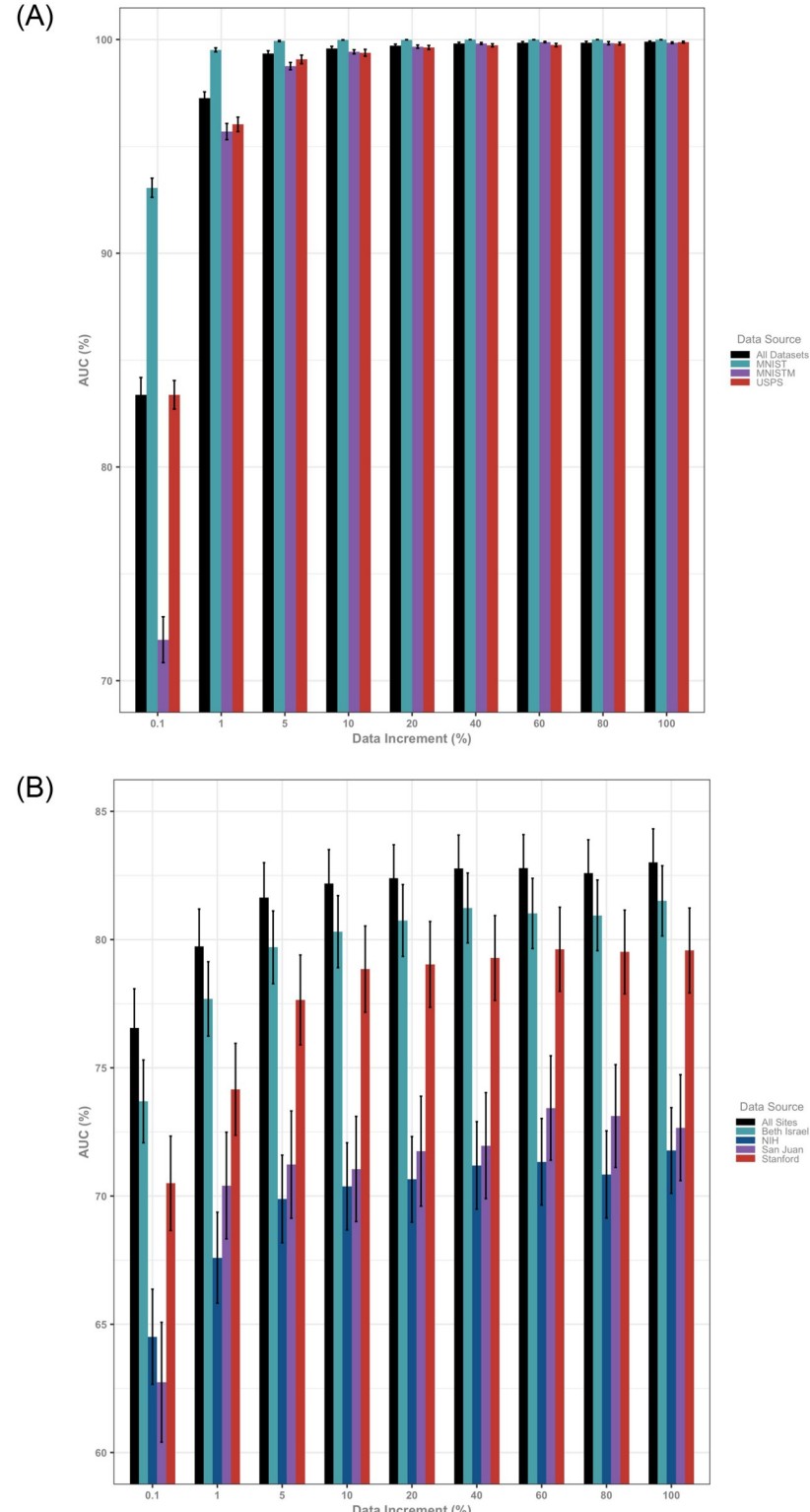

**Fig 4. Results of baseline global models trained on incremental amounts of available data and evaluated on the global test set and dataset-specific test sets demonstrate a discrepancy between global results and population (domain) specific results.** Error bars denote standard deviations. **(A)** Training and testing on an aggregate dataset obscures the fact that the model trained on all of the data has a difference in performance on digit classification of over 20% arguing against the practical utility of testing on aggregated data. This discrepancy is ameliorated by increasing

amounts of data and vanishes at 10% of the total available amount of data. **(B)** These results are initially mirrored in the chest x-ray cohort where performance of the global model trained on chest x-rays from all hospital sites and evaluated on the global and dataset-specific test sets demonstrates over 10% change in performance at 0.1% of the total available amount of data. Notably this discrepancy between site-specific performance is only mildly alleviated by increasing amounts of data and remains even when the joint model is trained on the entirety of the available dataset.

adaptation efficacy across three different digits datasets amounting to an average salvage of 35% AUC post-adaptation. We apply the algorithm to chest x-rays derived from four health-care institutions both in the United States and abroad, and show an improvement of 25% AUC in the identification of lung opacities relative to baseline. Finally, we show that the oft-used approach of creating a global model trained on data across domains is not sufficient to overcome data shift. Notably, we propose a unique metric, which we term domain spread, to characterize the heterogeneity of data across domains to obtain an a priori estimate of the external validity of a trained algorithm for any dataset. Taken together, these findings provide a comprehensive analysis of dataset shift, the potential of unsupervised domain adaptation, and a framework for future studies aiming to characterize and alleviate the burden of data shift in the widespread deployment of machine learning in the medical domain.

In many medical, financial, and military applications data is not portable, and it is necessary to find ways of training a model on locally available data that will be performant elsewhere. While there are novel collaborative means around this such as Federated Learning, these technologies still necessitate the labeling of target data and the collaboration of target sites with optimizing a local model on site [30]. Future research should continue to investigate computational methods for unsupervised domain adaptation to improve on our results. Furthermore, research into semi-supervised domain adaptation where limited out-of-population data can inform the transfer learning task may be extremely beneficial for improving performance on this problem. We note that by testing on population specific test sets, model performance directly comments on the nature of the underlying domain and its apparent learnability. We consider an encouraging future direction of research to involve investigating means of utilizing limited knowledge of the underlying domain to inform the domain adaptation task.

An important limitation of our study is label quality between the datasets, and we suspect that performance on the CXR task across all datasets is limited somewhat by the underlying quality of the labeling. We attempted to address this by utilizing the presence or absence of pulmonary "opacity" as our labels, and by noting that our key observation is the relative change in performance between internal, external, and adapted datasets. Furthermore, we recognize that training GANs is a computationally intensive task that requires hardware capabilities absent in many healthcare settings. We consider an important research direction to be one that reduces the computational burden of algorithm development in order to democratize the application of machine learning in the healthcare space.

## 4 Conclusion

We demonstrate that the use of unsupervised domain adaptation techniques can broadly increase model performance on external, shifted data. By measuring domain spread, we can determine a priori whether a global model provides a distinct advantage over domain-specific models. Improvements in domain adaptation such as shape-aware meta learning, and federated frequency attention maps may reduce the value of the domain spread, so domain spread can serve as an important marker for cross-site generalizability. Nevertheless, we anticipate that biases that exist within datasets such as those in label quality or underdiagnosis bias will need solutions that expand beyond purely computational approaches [31]. Future research

should investigate these biases and further utilize unsupervised learning as sparsely labeled datasets and high-quality, resource-intensive labelling become increasingly important [32]. When used in a challenging medical use case of practical importance—identification of lung opacity on CXRs—we note that adverserial domain adaptation leads to a generalized increase in performance. Purely computational approaches to handling dataset shift are not only tractable, but beneficial, and we believe will become an increasingly important part of the deployment toolkit for machine learning as these tools become increasingly used in medicine.

## 5 Methods

### 5.1 Datasets

Hand drawn digits were obtained from their standard online repositories for computer science research (S1 Table in S1 File). Further details and descriptions of these datasets are easily accessible online and we will omit further discussion of them. The clinical CXR data was obtained from four publicly-available retrospective datasets of chest x-rays, each containing images from thousands of patients, and summarized below (S2 Table in S1 File).

**5.1.1 ChestX-ray8 [22].** The ChestX-ray8 dataset derived from the National Institutes of Health contains 112,120 x-rays from 30,805 unique patients obtained over the period 1992–2015. Eight disease labels were extracted from radiology reports associated with each image: "atelectasis", "cardiomegaly", "effusion", "infiltration", "mass", "nodule", "pneumonia", and "pneumothorax". Scans without pathology were labeled "normal".

**5.1.2 CheXpert [21].** The CheXpert dataset derived from Stanford Hospital consists of 223,648 chest x-rays from 64,740 unique patients collected between October 2002 and July 2017. 14 labels were extracted from corresponding radiology reports: "atelectasis", "cardiomegaly", "consolidation", "edema", "enlarged cardiomegaly", "fracture", "lung lesion", "lung opacity", "no finding", "pleural effusion", "pleural other", "pneumonia", "pneumothorax", and "support devices".

**5.1.3 MIMIC-CXR [20].** The MIMIC-CXR dataset derived from Beth Israel Deaconess Medical Center in Boston consists of 371,920 chest x-rays from 65,088 unique patients obtained over the period 2011–2016. Labels in this study were identical to those used by CheXpert.

**5.1.4 PadChest [23].** The PadChest dataset derived from San Juan Hospital in Spain contains 160,861 chest x-rays from 67,625 patients between January 2009 and December 2017. 174 radiographic findings, 19 differential diagnoses, and 104 anatomic characteristics were extracted from radiology reports associated with each image.

### 5.2 Clinical taxonomy and pre-processing

**5.2.1 Taxonomy.** A significant challenge of working with multiple different clinical datasets for classification is heterogeneity in the labeling of the data. This is particularly notable for chest radiography where some datasets utilize language associated with radiographic findings (PadChest), while other datasets use mixed language that includes clinical diagnoses such as pneumonia. The challenge of label heterogeneity is compounded by error built into the labeling process itself, with studies utilizing a mix of manual annotation and semi-automated methods such as natural language processing to generate ground truth labels.

In order to obtain more homogenous labels and minimize class imbalance, we grouped together labels that were associated with opacities on AP CXR as indicated below. This single label ("opacity") was used for subsequent experiments. Use of this label is particularly relevant in the clinical setting because it allows one to capture the spectrum of radiographically visible pathology, which maximizes the intended utility of chest radiographs as front-line screening

tools in the clinical setting. It was derived by grouping pathologies as follows: *ChestX-ray8*: "atelectasis", "consolidation", "edema", "infiltration", "mass", "nodule", "pneumonia" *CheXpert/MIMIC-CXR*: "atelectasis", "consolidation", "edema", "lesion", "lung opacity", "pneumonia" *PadChest*: "alveolar pattern", "atelectasis", "atelectasis basal", "atypical pneumonia", "bronchiectasis", "calcified densities", "calcified granuloma", "calcified pleural plaques", "cavitation", "consolidation", "granuloma", "ground glass pattern", "increased density", "infiltrates", "interstitial pattern", "laminar atelectasis", "lobar atelectasis", "lung metastasis", "mass", "multiple nodules", "nodule", "pleural plaques", "pneumonia", "pseudonodule", "pulmonary edema", "pulmonary mass", "reticulonodular interstitial pattern", "round atelectasis", "segmental atelectasis", "soft tissue mass", "total atelectasis", "tuberculosis", "tuberculosis sequelae"

**5.2.2 Preprocessing.** Digits were nearest-neighbor interpolated to dimensions of 32x32x3. Chest x-rays were bilinear interpolated to dimensions of 224x224x3. Frontal chest x-rays were utilized for experimentation. Both digit and chest x-ray images were normalized with a mean and standard deviation of 0.5 for algorithm development.

### 5.3 Deep learning architectures

**5.3.1 Baseline CNN classification.** The LeNet architecture was used for baseline digit classification whereas the DenseNet architecture was chosen for chest x-ray opacity classification to maximize the ability of the network to learn fine-grained radiographic features. The ImageNet pretrained DenseNet feature extractor was concatenated to two fully connected layers composed of 1,000 and 100 hidden nodes interspersed with batch normalization, ReLU nonlinearity, and 50% dropout followed by a linear classifier. Hyper-parameters were selected by grid-search.

CNN models were trained on a single NVIDIA Tesla V100 using PyTorch 1.1.0. Models were trained using a categorical cross entropy loss with Adam optimizer. LeNet was trained with a learning rate of 0.001 and batch size 128, while DenseNet was trained with a learning rate of 0.0002, weight decay 0.0005, and batch size 50. Hyper-parameters were selected by grid search. Batches were balanced by opacity label for all chest x-ray experiments. Real-time affine data augmentation, including random flips, rotations, and translations, was conducted during DenseNet training. All models were trained for 200 epochs or early-stopped once validation AUC no longer improved for ten consecutive epochs.

A LeNet and DenseNet model was trained for each digit and chest x-ray dataset, respectively. Models were trained on an 80% split of a given dataset and validated on a 10% hold-out sample. The remaining 10% of data was used to construct a bootstrap sample of 1,000 replicates of size 1,000 to compute the ROC and other classification metrics.

**5.3.2 Adversarial domain adaptation implementation.** We adapt StarGAN for image-to-image translation [33]. The discriminator network uses a PatchGAN architecture, which classifies local MxM image patches as real or fake to promote fine-grained image synthesis. It is composed of six convolutional layers with kernel size four, stride two, and padding one interspersed with Leaky ReLU nonlinearity parametrized with a negative slope of 0.01 followed by two output convolutional layers. The generator network is composed of two downsampling convolutional layers with kernel size four, stride two, and padding one; residual blocks of size four and nine were used for digit and chest x-ray experiments, respectively; two transposed convolutional layers were used for upsampling. All convolutional layers were interspersed with Instance Normalization and ReLU nonlinearity.

The source task network adopts the LeNet or DenseNet architecture for digit and chest x-ray experiments, respectively. Initial training on source images and labels is as described in

'Baseline CNN classification'. The target task network was initialized with weights from the source network prior to subsequent training.

All models were trained using the Adam optimizer with $\beta_1 = 0.5$ and $\beta_2 = 0.999$ for 200K iterations. The learning rate was initialized at 0.0001 and linearly decayed toward zero after 100K training steps. Batches of size 200 and 20 were used for digit and chest x-ray experiments, respectively, with an even split of input images from the source and target domains. Batches were balanced by opacity label for all chest x-ray images derived from the source domain. Per StarGAN, the discriminator was updated five times for every generator update. The task networks were updated at every generator update, except in the MNISTM → USPS transformation where task networks were updated once every ten generator updates for stability of training.

Following training completion, train images from the source domain were transformed into synthetic images from the target domain. The transformed images were used to fine-tune the baseline CNN model (described in 'Baseline CNN classification'). The baseline and fine-tuned models were evaluated on test images from the target domain to evaluate efficacy of domain adaptation. All algorithm development was conducted on a single NVIDIA Tesla V100 using PyTorch 1.1.0.

**5.3.3 Adversarial domain adaptation design.** Adversarial modeling is a computational framework in which two algorithms are simultaneously trained in a minimax adversarial process. A simple instantiation of adversarial learning is the generative adversarial network (GAN), which is composed of a generative model G that approximates a data distribution pitted against a discriminative model D that aims to determine whether sample data is derived from the generative distribution or true data distribution. By way of analogy, G can be conceptualized as a counterfeiter trying to create fake currency that resembles the original, while D represents law enforcement trying to discern between fake and real currency. The adversarial process allows both algorithms to iteratively improve, ultimately allowing G to produce samples that are indistinguishable from genuine counterparts.

Numerous GANs have been proposed for a variety of computer vision tasks, including image synthesis and super-resolution imaging [33, 34]. The task of unsupervised domain adaptation, which aims to transfer insights gained from labeled data in a source domain in order to achieve comparable performance on unlabeled data in a target domain, has also seen applications of GANs. CyCADA, proposed by Hoffman et al., builds upon the GAN framework by introducing cycle and semantic consistency constraints to preserve pixel-level features and labels, respectively, when mapping from source to target domains [35]. The formulation requires two pairs of generators and discriminators to map across domains and achieves state-of-the-art performance on digit classification. StarGAN, proposed by Choi et al., incorporates the cycle consistency constraint and scales domain adaptation to multiple domains using a single generator and discriminator framework by conditioning the generator on a target domain label. It achieves among the best results on facial attribute transfer and facial expression synthesis [33].

*Problem formulation.* This work focuses on the task of unsupervised domain adaptation, where given source data $X_S$, source labels $Y_S$, and target data $X_C$ but no target labels $Y_C$, the goal is to learn a classification model $F$ that can correctly predict the label for $X_C$. In this case, we assign $C$ to refer the the target domain. A naive approach would train a classification algorithm on source data and apply it to predict labels for data in the target domain. However, such an approach has shown to exhibit diminishing performance due to domain shift across data domains [36–40]. StarGAN is a novel GAN framework that extends upon previous methods to map data across multiple domains while preserving fine-grained image and semantic characteristics for application in the medical context [33].

*Image-to-image translation.* Given $X_S$ and $X_C$, we adopt StarGAN to learn a mapping of domains $S \rightarrow C$ and $C \rightarrow S$ using a single generator $G$ such that discriminator $D$ is unable to distinguish real and synthetic images across domains. In other words, we want the generator, given a sample and a target domain to map to an image in the sample domain ($x_s$) to an equivalent image in the the target domain ($x \rightarrow c$) $G(x_s, c) \rightarrow x_c$. Our discriminator, on the other hand, produces a probability distribution over both the source sources and domain labels. $D$: $x \rightarrow \{P_s(x), P_c(x)\}$, where $P_s$ and $P_c$ is the probability that the sample belongs in the source or target domain, respectively. Per StarGAN, the objective expressed as the adversarial loss

$$L_{adv}(D, G, X, C) = E_{x \sim X}[\log D_s rc(x)] + E_{(x,c) \sim (X,C)}[\log(1 - D_{src}(G(x, c)))] \tag{1}$$

enables $G$ to generate an image $G(x, c)$ that is indistinguishable from real images. D is the discriminator network, $G$ is the generator network, $X$ is the set of real samples, $Y$ is the set of domains associated with $X$. $(x, c)$ are a pair where x refers to a sample from the data source, and $c$ refers to the target domain label.

To stabilize adversarial training, we adopt the Wasserstein GAN objective from StarGAN defined as

$$L_{adv}(D, G, X, C) = E_{x \sim X}[D_{src}(x)] - E_{(x,c) \sim (X,C)}[D_{src}(G(x, c))] - \lambda_{gp} E_{\hat{x} \sim P\hat{x}}\left[(||\nabla \hat{x} D(\hat{x})||_2 - 1)^2\right] \tag{2}$$

where $P\hat{x}$ is derived from uniformly sampling along straight lines between coupled points from the true data and generator distributions. $E_{(x \sim X)}[D_{src}(x)]$ is the expected loss of the discriminator on the source data, $E_{(x,c) \sim (X, C)}[D_{src}(G(x, c))]$ is the expected loss of the discriminator of the source data on generated data in the target domain, and $\lambda_{gp} E_{\hat{x} \sim P\hat{x}}[(||\nabla \hat{x} D(\hat{x})||_2 - 1)^2]^2]$ is a regularization term to minimize the gradient as described in [41]. All experiments use $\lambda_g p = 10$, which was optimized via grid-search. As in StarGAN, to constrain G to produce images in the target domain c, a domain classification loss is also imposed on both D and G. A simple classification loss over real images ($L^r_{classification}$) is used to optimize $D$

$$L^r_{classification}(D, X, C) = E_{(x,C) \sim (X,C)}[NLL(D(x), c)] \tag{3}$$

Where, $c$ is the domain of the sample $x$, and $NLL(D(x), c)$, is a negative log-likelihood loss of the discriminator $D(x)$, which predicts a given class, and the true class c. Conversely, the classification of D with respect to the fake images generated by the generator network $G(x, c)$ is used to optimize $G$, as is standard for generative adversarial networks.

$$L^f_{cls}(D, G, X, C) = E_{(x,c) \sim (X,C)}[NLL(D(G(x, c)), c)] \tag{4}$$

By minimizing this objective, G is able to generate images that can be classified as target domain c. Although the adversarial and domain classification losses constrain G to generate images that appear realistic in the target domain, they do not preserve the content of the input image independent from domain characteristics. Therefore, a $L_1$ penalty on the cycle consistency loss as defined in StarGAN.

$$L_{cyc}(G, X, C) = E_{(x,c,c') \sim (X,C,C')}[||x - G(G(x, c), c')||_1] \tag{5}$$

is imposed on the generator such that G is constrained to reconstruct input x in the original domain c' from the translated image $G(x, c)$. $[||x - G(G(x, c), c')||_1$ is a reconstruction loss with an absolute-value based normalization.

*Semantic modeling.* Although the image translation framework enables G to synthesize realistic images in a given domain, it does not guarantee the preservation of semantic information

across domains. For example, when translating chest x-rays from source to target domain, $G$ may not maintain the underlying disease content. Therefore, we adopt techniques from CYCADA to train a source classifier $F_S$ to weakly supervise $G$ to generate images that are classified the same way before and after translation. Images generated in the target domain that correspond to the source domain should have similar labels. In other words, we use the labels that have already been prescribed in the source domain to guide the generator $G$. First, a classifier ($F_s$) is trained on the labeled source data with cross entropy loss.

$$L_{task}(F_S, X_S, Y_S) = -E_{(x_s,y_s)\sim(X_S,Y_S)}\sum_{n=1}^{N}1_{[n=y_s]}\log(\sigma(F_S^{(n)}(x_s))) \quad (6)$$

where $N$ denotes the number of classes and $\sigma$ is the softmax function.

Second, a target classifier $F_T$ is fine-tuned from $F_S$ to weakly supervise G on fake images in the target domain. Ablation studies from the CYCADA paper have shown that this step leads to improvements in domain adaptation. Subsequently, the semantic loss—where the goal is the maintain a semantic relationship between the images in the target domain and the source domain can be defined with respect to $F_S$ and $F_T$. This loss is simply the addition of for the classifier for images from the source domain $L_{task}(F_S, G(G(X_S, C), C'), Y_S$, the classifier for images from the target domain ($L_{task}(F_T, G(X_S, C), Y_S)$ and the classifier for the images from the source domain with respect to the predictions generated by the classifier of images in the target domain $L_{task}(F_S, G(X_T, C), F_T(X_T))$.

$$L_{sem}(G, F_S, F_T, X_S, Y_S, X_T, C) = L_{task}(F_T, G(X_S, C), Y_S)$$

$$+L_{task}(F_S, G(G(X_S, C), C'), Y_S) + L_{task}(F_S, G(X_T, C), F_T(X_T)) \quad (7)$$

By generating a loss function that combines all three aspects, can generate a semantic relationship between the classifications generated by the source samples and the classifications of samples in target domain.

*Final Objective* Taken together, the objective functions to optimize D and G, respectively, are:

$$L_D = -L_{adv} + \lambda_{cls}L_{cls}^r \quad (8)$$

$$L_G = L_{adv} + \lambda_{cls}L_{cls}^f + \lambda_{cyc}L_{cyc} + L_{sem} \quad (9)$$

where $\lambda_{cls}$ and $\lambda_{cyc}$ are scalars that control the importance of domain classification and cycle consistency losses, respectively. As per StarGAN, $\lambda_{cls} = 1$ and $\lambda_{cyc} = 10$ for all experiments.

## 5.4 Statistical analysis

Dataset characteristics were compared using Analysis of Variance (ANOVA) where appropriate. Bootstrap confidence intervals were constructed to compare results at baseline and post-adaptation using bootstrap samples of 1,000 replicates. Statistical significance was evaluated at an alpha level of 0.05. All statistical analyses were performed in Python 3.5.2.

## Supporting information

**S1 File.**
(PDF)

## Acknowledgments

The authors thank Amy Zhong, MA for her work on the illustrations in this study.

## Author Contributions

**Conceptualization:** Aly A. Valliani, Chiatse Wang, Weichung Wang, Anthony B. Costa, Eric K. Oermann.

**Data curation:** Aly A. Valliani.

**Formal analysis:** Aly A. Valliani, Eric K. Oermann.

**Investigation:** Aly A. Valliani, Eric K. Oermann.

**Methodology:** Aly A. Valliani, Eric K. Oermann.

**Project administration:** Eric K. Oermann.

**Resources:** Aly A. Valliani, Anthony B. Costa, Eric K. Oermann.

**Software:** Aly A. Valliani, Anthony B. Costa, Eric K. Oermann.

**Supervision:** Douglas Kondziolka, Eric K. Oermann.

**Validation:** Aly A. Valliani, Eric K. Oermann.

**Visualization:** Aly A. Valliani, Michael L. Martini, Eric K. Oermann.

**Writing – original draft:** Aly A. Valliani, Michael L. Martini, Eric K. Oermann.

**Writing – review & editing:** Aly A. Valliani, Faris F. Gulamali, Young Joon Kwon, Michael L. Martini, Chiatse Wang, Douglas Kondziolka, Viola J. Chen, Weichung Wang, Anthony B. Costa, Eric K. Oermann.

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
