## [Decision Letter · Decision Letter 0]

9 May 2022

PONE-D-22-08145Deploying deep learning models on unseen medical imaging using adversarial domain adaptationPLOS ONE

Dear Dr. Oermann,

Thank you for submitting your manuscript to PLOS ONE. After careful consideration, we feel that it has merit but does not fully meet PLOS ONE’s publication criteria as it currently stands. Therefore, we invite you to submit a revised version of the manuscript that addresses the points raised during the review process.

We look forward to receiving your revised manuscript.

Kind regards,

Mohamed Hammad, Ph.D.

Academic Editor

PLOS ONE

Journal Requirements:

2. Please note that PLOS ONE has specific guidelines on code sharing for submissions in which author-generated code underpins the findings in the manuscript. In these cases, all author-generated code must be made available without restrictions upon publication of the work. Please review our guidelines at https://journals.plos.org/plosone/s/materials-and-software-sharing#loc-sharing-code and ensure that your code is shared in a way that follows best practice and facilitates reproducibility and reuse. Code may be shared by providing a URL within the Methods section to a code repository or it may be uploaded as a supplemental file.

Reviewers' comments:

Reviewer's Responses to Questions

**Comments to the Author**

1. Is the manuscript technically sound, and do the data support the conclusions?

Reviewer #1: No

Reviewer #2: Partly

2. Has the statistical analysis been performed appropriately and rigorously? 

Reviewer #1: No

Reviewer #2: No

3. Have the authors made all data underlying the findings in their manuscript fully available?

Reviewer #1: Yes

Reviewer #2: No

4. Is the manuscript presented in an intelligible fashion and written in standard English?

Reviewer #1: No

Reviewer #2: No

5. Review Comments to the Author

Reviewer #1: The paper has been carelessly prepared and needs to be largely reworked. These are some points that the authors must particularly pay attention and handle:

• The major problem of this work is that its novelty and the theoretical contribution are so limited. So, the authors should modify it carefully and improve the novelty of this paper. Also, the authors should provide solid motivation for their work based on the existing literature.

• The figures need to be amended. For example, the font is too small, and the resolution is not clear which makes it difficult to read.

• All the equations were missing.

• Please add Figure or Table about the optimal structure of the proposed method. In addition, provide the values of all parameters of proposed method in table.

• Please specify how the parameters of proposed method were selected.

• Please specify if the proposed methods parameters were optimized. If so, please write how proposed parameters were optimized?

•

• More details about the simulation software exploited should be added.

• The results should be further analyzed, more details and further discussion of the simulation results is needed.

• I recommend the authors to review below works and incorporate them while revising the paper:

1. Zhang, Ling, et al. "Generalizing deep learning for medical image segmentation to unseen domains via deep stacked transformation." IEEE transactions on medical imaging 39.7 (2020): 2531-2540.

2. Yin, Baocai, et al. "AFA: adversarial frequency alignment for domain generalized lung nodule detection." Neural Computing and Applications (2022): 1-12.

3. Liu, Quande, Qi Dou, and Pheng-Ann Heng. "Shape-aware meta-learning for generalizing prostate MRI segmentation to unseen domains." International Conference on Medical Image Computing and Computer-Assisted Intervention. Springer, Cham, 2020.

4. Liu, Quande, et al. "Feddg: Federated domain generalization on medical image segmentation via episodic learning in continuous frequency space." Proceedings of the IEEE/CVF Conference on Computer Vision and Pattern Recognition. 2021.

• The conclusions section should conclude that you have achieved from the study, contributions of the study to academics and practices. In addition, list the advantages and disadvantages of the proposed solution, as well as indicate the limitations of work. Further, mention the recommendations of future works.

• The list of references should be reformatted and checked again to be matched with the journal requirement where a different styles and types are used.

Reviewer #2: In this manuscript, the authors develop a general technique for ameliorating the effect of dataset shift using generative adversarial networks (GANs) on a dataset of 149,298 handwritten digits and a dataset of 868,549 chest radiographs obtained from four academic medical centers. They assess efficacy by comparing the area under the curve (AUC) pre-and post-adaptation. Adversarial domain adaptation leads to improved model performance on radiographic data derived from multiple out-of-sample healthcare populations. Their work can be applied to other medical imaging domains to help shape the deployment toolkit of machine learning in medicine.

Before its acceptance for publication, the authors must arrange all the proposed models to be readable. Indeed, from line 357 to line 408, all equations are not readable. Also, the presentation of the manuscript must be improved.

6. PLOS authors have the option to publish the peer review history of their article (what does this mean?). If published, this will include your full peer review and any attached files.

Reviewer #1: No

Reviewer #2: No

---

## [Author Response · Author response to Decision Letter 0]

11 Jul 2022

1 June 2022

To Dr. Mohamed Hammad

Thank you for your consideration of our manuscript. We have addressed all the comments provided, which we think have certainly improved the manuscript. Below, we provide the original comments and include our point-by-point responses.

Comments:

Reviewer #1:

The major problem of this work is that its novelty and the theoretical contribution are so limited. So, the authors should modify it carefully and improve the novelty of this paper. Also, the authors should provide solid motivation for their work based on the existing literature.

We thank the reviewer for the comment. The novelty of this paper arises from its technical extension of existing domain adaptation approaches but more importantly its extensive application of an algorithm toward general amelioration of dataset shift and characterization of domain spread as an a priori estimate of the prevalence of domain shift in a given dataset. Further, we contextualize our paper within these papers with the following paragraph included in the introduction, and cite the papers referenced as recommended by the reviewer.

“Prior work has utilized variations of adversarial domain adaptation on a spectrum of different tasks including medical image segmentation, lung nodule detection, prostate MRI segmentation, and federated learning. For example, previous methods have trained on augmented big data in the domains of prostate, left atrial, and left ventricular and shown that augmentation reduces the degradation in performance significantly.(14) In our study, we primarily focus on an in-hospital vs out-of-hospital cohort, rather than differing tasks altogether. A second method has utilized adaptive transition module (ATM) to learn a frequency attention map that can align different domain images in a common frequency domain. By backpropagating with differentiable fast fourier transform, lung nodule detection performance was significantly improved.(15) We do not use a frequency domain, but we anticipate that applying a frequency-based normalization may also improve performance. Shape-aware meta learning utilizes a network that can learn shape compactness and shape smoothness to provide domain-invariant embeddings.(16) Similar to ATMs, shape-aware meta-learning is primarily focused on different objectives rather than learning out-of-sample embeddings. Finally, some methods are able to combine Fourier transforms and shape-aware meta learning, demonstrating improved performance on out-of-sample objectives.(17) In context, our paper focuses on investigating the a priori assumption of dataset shift, and how it can be utilized to improve performance across centers rather than generating a novel machine learning methods to combat domain shift.”

14. Zhang, Ling, et al. "Generalizing deep learning for medical image segmentation to unseen domains via deep stacked transformation." IEEE transactions on medical imaging 39.7 (2020): 2531-2540.

15. Yin, Baocai, et al. "AFA: adversarial frequency alignment for domain generalized lung nodule detection." Neural Computing and Applications (2022): 1-12.

16. Liu, Quande, Qi Dou, and Pheng-Ann Heng. "Shape-aware meta-learning for generalizing prostate MRI segmentation to unseen domains." International Conference on Medical Image Computing and Computer-Assisted Intervention. Springer, Cham, 2020.

17. Liu, Quande, et al. "Feddg: Federated domain generalization on medical image segmentation via episodic learning in continuous frequency space." Proceedings of the IEEE/CVF Conference on Computer Vision and Pattern Recognition. 2021.

The figures need to be amended. For example, the font is too small, and the resolution is not clear which makes it difficult to read.

Thank you for the feedback. We have provided figures at the highest vectorized resolution. We are happy to reformat figures further if zooming does not enable adequate viewing.

All the equations are missing.

Thank you for the comment. It appears the formulas were accidentally redacted upon initial submission. They have now been incorporated with additional details. The summary equations of the generator and the discriminator are listed in lines 442 and 443. Equations specific to the StarGAN are listed in line 404, 408, 413, 416, and 423. The task and semantic loss are defined in line 433 and 439. 

Please add Figure or Table about the optimal structure of the proposed method. 

Thank you for the comment. Given the various training regimes utilized, we have provided Figure 1C as a generalizable illustration of the algorithm used across experiments. The authors believe creating an exhaustive table of all parameters would be unwieldy (especially given they were not custom tuned as mentioned in the next comment). However, in order to add further clarity, we have made the equations more explainable by adding thorough descriptions for each in the text.

For example for equation 1, we have added. 

In other words, we want the generator, given a sample and a target domain to map to an image in the sample domain (xs) to an equivalent image in the the target domain (x¬c) G(x_s,c)→x_c. Our discriminator, on the other hand, produces a probability distribution over both the source sources and domain labels. D:x→{P_s (x),P_c (x)}, where P¬s and Pc is the probability that the sample belongs in the source or target domain, respectively

For equation 2, we have added 

“E_(x～X) [D_src (x)] is the expected loss of the discriminator on the source data, E_((x,c)～(X,C)) [D_src (G(x,c))] is the expected loss of the discriminator of the source data on generated data in the target domain, and λ_gp 〖E_x ^ 〗_(～P_x ^ ) [(||∇_x ^ D(x ^)||_2-1)^2] is a regularization term to minimize the gradient as described in 41. All experiments use ƛgp = 10, which was optimized via grid-search.”

For equation 3 and 4, we have simplified the equation to the negative log-likelihood loss.

. A simple classification loss over real images (〖L^r〗_classification) is used to optimize D

〖L^r〗_classification (D,X,C)= E_((x,)～(X,) ) [NLL(D(x),c)] (3)

Where, c is the domain of the sample x, and NLL(D(x),c), is a negative log-likelihood loss of the discriminator D(x), which predicts a given class, and the true class c. Conversely, the classification of D with respect to the fake images generated by the generator network G(x,c) is used to optimize G, as is standard for generative adversarial networks. 

 〖L^f〗_cls (D,G,X,C)=E_((x,c)～(X,C) ) [NLL(D(G(x,c)),c) ] (4)

For equation 5, we have defined the L1 loss

[||x - G(G(x,c),c')||_1 is a reconstruction loss with an absolute-value based normalization. 

For equation 6, we provide the definition of semantic similarity to clarify the meaning behind the equations. 

Images generated in the target domain that correspond to the source domain should have similar labels. In other words, we use the labels that have already been prescribed in the source domain to guide the generator G. First, a classifier (F¬s) is trained on the labeled source data with cross entropy loss. 

For equation 7, we explain the reasoning behind the additive losses. 

Ablation studies from the CYCADA paper have shown that this step leads to improvements in domain adaptation. Subsequently, the semantic loss – where the goal is the maintain a semantic relationship between the images in the target domain and the source domain can be defined with respect to FS and FT. This loss is simply the addition of for the classifier for images from the source domain (L_task (F_S,G(G(X_S,C),C'),Y_S), the classifier for images from the target domain (L_task (F_T,G(X_S,C),Y_S ) and the classifier for the images from the source domain with respect to the predictions generated by the classifier of images in the target domain L_task (F_S,G(X_T,C),F_T (X_T)). 

L_sem (G,F_S,F_T,X_S,Y_S,X_T,C)=L_task (F_T,G(X_S,C),Y_S)+L_task (F_S,G(G(X_S,C),C'),Y_S)

 + L_task (F_S,G(X_T,C),F_T (X_T)) (7)

By generating a loss function that combines all three aspects, can generate a semantic relationship between the classifications generated by the source samples and the classifications of samples in target domain. 

In addition, provide the values of all parameters of proposed method in table. Please specify how the parameters of proposed method were selected. Please specify if the proposed methods parameters were optimized. If so, please write how proposed parameters were optimized?

Thank you for this important point of clarification. Truthfully, we did not use a systematic method for hyperparameter tuning for three reasons: 1) GAN training is often extremely unstable and there are few (if any) a prior hyperparameter settings to narrow the search space, 2) each experiment required many hours to conduct given the extent of our dataset and the number of adaptation pairs simulated which made a broad hyperparameter search unfeasible, and 3) our focus was not to overoptimize our algorithm but instead to provide a generalizable approach for domain adaptation among diverse adaptation tasks (digits and CXRs) and adaptation pairs. As such, we leveraged insights raised by members of the team given our prior experience applying GANs to medical image data to conduct targeted adjustments of hyperparameter settings. Many of the parameters leveraged are consistent with those in the literature. For example, the beta_1 value of 0.5 was the optimized value used by authors of the StarGAN paper upon which our algorithm is built. This value has been shown in prior work (Radford et al. ICLR 2016: Unsupervised representation learning with deep convolutional generative adversarial networks) to stabilize generator training and has been adopted in other work as well (Gulrajani et al. NIPS 2017: Improved training of Wasserstein GANs).

More details about the simulation software exploited should be added.

Thank you for this point of clarification. No simulation software was utilized in this study. Figure 1 details the training regime whereby images from a single dataset (or hospital in the case of chest radiographs) were adapted to mirror that from another dataset or hospital, which resulted in improved classification efficacy. We are happy to provide additional details as requested.

The results should be further analyzed, more details and further discussion of the simulation results is needed.

Thank you for this request. We have included the following in the results for digit adaptation:

“For example, adaptations between MNIST and MNISTM yielded significant improvements in classification performance due to the similar baseline character style across the two datasets. Adaptations between MNIST and USPS were similarly efficacious due to transition across grayscale domains whereas adaptations between MNISTM and USPS were less successful given the more difficult task of adaptation across character styles and color domains.”

I recommend the authors to review below works and incorporate them while revising the paper:

15. Zhang, Ling, et al. "Generalizing deep learning for medical image segmentation to unseen domains via deep stacked transformation." IEEE transactions on medical imaging 39.7 (2020): 2531-2540.

16. Yin, Baocai, et al. "AFA: adversarial frequency alignment for domain generalized lung nodule detection." Neural Computing and Applications (2022): 1-12.

17. Liu, Quande, Qi Dou, and Pheng-Ann Heng. "Shape-aware meta-learning for generalizing prostate MRI segmentation to unseen domains." International Conference on Medical Image Computing and Computer-Assisted Intervention. Springer, Cham, 2020.

18. Liu, Quande, et al. "Feddg: Federated domain generalization on medical image segmentation via episodic learning in continuous frequency space." Proceedings of the IEEE/CVF Conference on Computer Vision and Pattern Recognition. 2021.

We enjoyed reviewing these manuscripts and have included them into the paragraph per our response to the first comment.

The conclusions section should conclude that you have achieved from the study, contributions of the study to academics and practices. In addition, list the advantages and disadvantages of the proposed solution, as well as indicate the limitations of work. Further, mention the recommendations of future works.

Thank you for the comment. For the advantages, we have included, “By measuring domain spread, we can determine a priori whether a global model provides a distinct advantage over domain-specific models. Improvements in domain adaptation such as shape-aware meta learning, and federated frequency attention maps may reduce the value of the domain spread, so domain spread can serve as an important marker for cross-site generalizability.”

For the disadvantages, “Nevertheless, we anticipate that biases that exist within datasets such as those in label quality or underdiagnosis bias will need solutions that expand beyond purely computational approaches.” For recommendations regarding future work, we have included “Future research should investigate these biases and further utilize unsupervised learning as sparsely labeled datasets and high-quality, resource-intensive labelling become increasingly important.”

The list of references should be reformatted and checked again to be matched with the journal requirement where a different styles and types are used.

Thank you for the comment. We have verified that the citations are in an appropriate (Vancouver) citation style. 

Reviewer #2:

In this manuscript, the authors develop a general technique for ameliorating the effect of dataset shift using generative adversarial networks (GANs) on a dataset of 149,298 handwritten digits and a dataset of 868,549 chest radiographs obtained from four academic medical centers. They assess efficacy by comparing the area under the curve (AUC) pre-and post-adaptation. Adversarial domain adaptation leads to improved model performance on radiographic data derived from multiple out-of-sample healthcare populations. Their work can be applied to other medical imaging domains to help shape the deployment toolkit of machine learning in medicine.

Before its acceptance for publication, the authors must arrange all the proposed models to be readable. Indeed, from line 357 to line 408, all equations are not readable. Also, the presentation of the manuscript must be improved.

Thank you for noting the absence of equations and lack of clarity. It appears the formulas were accidentally redacted upon initial submission. Given the various training regimes utilized, we have provided Figure 1C as a generalizable illustration of the algorithm used across experiments. We did not use a systematic method for hyperparameter tuning for three reasons: 1) GAN training is often extremely unstable and there are few (if any) a prior hyperparameter settings to narrow the search space, 2) each experiment required many hours to conduct given the extent of our dataset and the number of adaptation pairs simulated which made a broad hyperparameter search unfeasible, and 3) our focus was not to overoptimize our algorithm but instead to provide a generalizable approach for domain adaptation among diverse adaptation tasks (digits and CXRs) and adaptation pairs. As such, we leveraged insights raised by members of the team given our prior experience applying GANs to medical image data to conduct targeted adjustments of hyperparameter settings. Many of the parameters leveraged are consistent with those in the literature. For example, the beta_1 value of 0.5 was the optimized value used by authors of the StarGAN paper upon which our algorithm is built. This value has been shown in prior work (Radford et al. ICLR 2016: Unsupervised representation learning with deep convolutional generative adversarial networks) to stabilize generator training and has been adopted in other work as well (Gulrajani et al. NIPS 2017: Improved training of Wasserstein GANs).

In order to add further clarity, we have made the equations more explainable by adding thorough descriptions for each in the text.

For example for equation 1, we have added. 

In other words, we want the generator, given a sample and a target domain to map to an image in the sample domain (xs) to an equivalent image in the the target domain (x¬c) G(x_s,c)→x_c. Our discriminator, on the other hand, produces a probability distribution over both the source sources and domain labels. D:x→{P_s (x),P_c (x)}, where P¬s and Pc is the probability that the sample belongs in the source or target domain, respectively

For equation 2, we have added 

“E_(x～X) [D_src (x)] is the expected loss of the discriminator on the source data, E_((x,c)～(X,C)) [D_src (G(x,c))] is the expected loss of the discriminator of the source data on generated data in the target domain, and λ_gp 〖E_x ^ 〗_(～P_x ^ ) [(||∇_x ^ D(x ^)||_2-1)^2] is a regularization term to minimize the gradient as described in 41. All experiments use ƛgp = 10, which was optimized via grid-search.”

For equation 3 and 4, we have simplified the equation to the negative log-likelihood loss.

. A simple classification loss over real images (〖L^r〗_classification) is used to optimize D

〖L^r〗_classification (D,X,C)= E_((x,)～(X,) ) [NLL(D(x),c)] (3)

Where, c is the domain of the sample x, and NLL(D(x),c), is a negative log-likelihood loss of the discriminator D(x), which predicts a given class, and the true class c. Conversely, the classification of D with respect to the fake images generated by the generator network G(x,c) is used to optimize G, as is standard for generative adversarial networks. 

 〖L^f〗_cls (D,G,X,C)=E_((x,c)～(X,C) ) [NLL(D(G(x,c)),c) ] (4)

For equation 5, we have defined the L1 loss

[||x - G(G(x,c),c')||_1 is a reconstruction loss with an absolute-value based normalization. 

For equation 6, we provide the definition of semantic similarity to clarify the meaning behind the equations. 

Images generated in the target domain that correspond to the source domain should have similar labels. In other words, we use the labels that have already been prescribed in the source domain to guide the generator G. First, a classifier (F¬s) is trained on the labeled source data with cross entropy loss. 

For equation 7, we explain the reasoning behind the additive losses. 

Ablation studies from the CYCADA paper have shown that this step leads to improvements in domain adaptation. Subsequently, the semantic loss – where the goal is the maintain a semantic relationship between the images in the target domain and the source domain can be defined with respect to FS and FT. This loss is simply the addition of for the classifier for images from the source domain (L_task (F_S,G(G(X_S,C),C'),Y_S), the classifier for images from the target domain (L_task (F_T,G(X_S,C),Y_S ) and the classifier for the images from the source domain with respect to the predictions generated by the classifier of images in the target domain L_task (F_S,G(X_T,C),F_T (X_T)). 

L_sem (G,F_S,F_T,X_S,Y_S,X_T,C)=L_task (F_T,G(X_S,C),Y_S)+L_task (F_S,G(G(X_S,C),C'),Y_S)

 + L_task (F_S,G(X_T,C),F_T (X_T)) (7)

By generating a loss function that combines all three aspects, can generate a semantic relationship between the classifications generated by the source samples and the classifications of samples in target domain.

---

## [Decision Letter · Decision Letter 1]

18 Jul 2022

PONE-D-22-08145R1Deploying deep learning models on unseen medical imaging using adversarial domain adaptationPLOS ONE

Dear Dr. Oermann,

Thank you for submitting your manuscript to PLOS ONE. After careful consideration, we feel that it has merit but does not fully meet PLOS ONE’s publication criteria as it currently stands. Therefore, we invite you to submit a revised version of the manuscript that addresses the points raised during the review process.

We look forward to receiving your revised manuscript.

Kind regards,

Mohamed Hammad, Ph.D.

Academic Editor

PLOS ONE

Journal Requirements:

Reviewers' comments:

Reviewer's Responses to Questions

**Comments to the Author**

1. If the authors have adequately addressed your comments raised in a previous round of review and you feel that this manuscript is now acceptable for publication, you may indicate that here to bypass the “Comments to the Author” section, enter your conflict of interest statement in the “Confidential to Editor” section, and submit your "Accept" recommendation.

Reviewer #1: All comments have been addressed

Reviewer #2: All comments have been addressed

2. Is the manuscript technically sound, and do the data support the conclusions?

Reviewer #1: Yes

Reviewer #2: Yes

3. Has the statistical analysis been performed appropriately and rigorously? 

Reviewer #1: Yes

Reviewer #2: Yes

4. Have the authors made all data underlying the findings in their manuscript fully available?

Reviewer #1: Yes

Reviewer #2: Yes

5. Is the manuscript presented in an intelligible fashion and written in standard English?

Reviewer #1: Yes

Reviewer #2: Yes

6. Review Comments to the Author

Reviewer #1: Since the previous version, authors have done huge work and the paper is much better. This version looks good Therefore, I suggest accepting this paper after minor:

• The figures still need to be amended, where the font is too small which makes it difficult to read.

Reviewer #2: The authors addressed all my comments in this version of the manuscript which was well improved. I recommend it for publication

7. PLOS authors have the option to publish the peer review history of their article (what does this mean?). If published, this will include your full peer review and any attached files.

Reviewer #1: No

Reviewer #2: No

---

## [Author Response · Author response to Decision Letter 1]

1 Aug 2022

30 July 2022

To Dr. Mohamed Hammad

Thank you for your consideration of our manuscript. We have addressed all the comments provided. Below, we provide the original comments and include our point-by-point responses.

Comments:

Reviewer #1:

Since the previous version, authors have done huge work and the paper is much better. This version looks good. Therefore, I suggest accepting this paper after minor: The figures still need to be amended, where the font is too small which makes it difficult to read.

We thank the reviewer for providing comments that have improved the manuscript. To improve readability, we have provided elements of each panel as individual figures, which are now of sufficient font size to ensure all readers may view without difficulty. This will also allow the journal editing team to format the figures as they deem best in the published form of the manuscript.

Reviewer #2:

The authors addressed all my comments in this version of the manuscript which was well improved. I recommend it for publication.

We thank the reviewer for taking the time to provide comments that have certainly improved the manuscript.

---

## [Editor Report · Decision Letter 2]

5 Aug 2022

Deploying deep learning models on unseen medical imaging using adversarial domain adaptation

PONE-D-22-08145R2

Dear Dr. Oermann,

We’re pleased to inform you that your manuscript has been judged scientifically suitable for publication and will be formally accepted for publication once it meets all outstanding technical requirements.

Kind regards,

Mohamed Hammad, Ph.D.

Academic Editor

PLOS ONE

---

## [Editor Report · Acceptance letter]

29 Sep 2022

PONE-D-22-08145R2 

Deploying deep learning models on unseen medical imaging using adversarial domain adaptation 

Dear Dr. Oermann:

I'm pleased to inform you that your manuscript has been deemed suitable for publication in PLOS ONE. Congratulations! Your manuscript is now with our production department. 

Kind regards, 

on behalf of

Dr. Mohamed Hammad 

Academic Editor

PLOS ONE